# COARSE TO FINE: UNCOVERING NODE DIVERSITY IN WEAKLY SUPERVISED GRAPH ANOMALY DETECTION

## ABSTRACT

Graph anomaly detection (GAD) aims to identify abnormal nodes in graph datasets, which is a significant and challenging task. Most existing methods regard the problem as a binary classification task when exploiting the labeled data, overlooking the potential existence of fine-grained subcategories among both normal and anomalous nodes. The coarse-grained treatment often results in a sub-optimal decision boundary, and the scarcity of labeled data makes it worse. To tackle these limitations, we propose a novel framework for GAD under weak supervision, addressing the problem via two key innovations. First, we introduce a unified gating module to tackle the diversity of anomaly types. It adaptively balances node-centric attributes and neighborhood signals within a single model, allowing it to identify different anomalous patterns like contextual and structural anomalies. Second, a classifier-clustering synergy framework is developed, under which the discovery of node sub-categories and the classification of anomalies can mutually reinforce each other. We achieve this by dynamically maintaining two high-confidence sets of normal and abnormal nodes, which are determined by both of the classifier and clustering modules. Extensive experiments on seven public graph datasets demonstrate that our method consistently outperforms existing approaches, validating its effectiveness in weakly supervised graph anomaly detection.

## 1 INTRODUCTION

Graphs are widely used data structures in real-life applications, including social networks, financial transaction networks, and E-commerce review systems. Graph anomaly detection (GAD) aims to identify anomalies that deviate significantly from normal patterns, such as spam reviews (McAuley & Leskovec, 2013), social bots (Yang et al., 2024), and financial fraud (Liu et al., 2024). These anomalies are prone to have a severe negative impact on platform security or user experience. Consequently, GAD has attracted widespread attention and research in various domains (Ding et al., 2019; Bandyopadhyay et al., 2020; Fan et al., 2020; Peng et al., 2020; Yuan et al., 2021; He et al., 2024; Roy et al., 2024). From the perspective of supervision, current GAD methods can be grouped into two categories—unsupervised learning and (semi-)supervised learning. Most unsupervised GAD methods are based on prior knowledge or assumptions about the deviation between normal and abnormal nodes when reconstruction (Ding et al., 2019; Yuan et al., 2021; Roy et al., 2024; He et al., 2024; Xi et al., 2024) or contrastive learning (Liu et al., 2021b; Chen et al., 2024a) is applied. Considering that it is often feasible to collect a set of labeled data, many semi-supervised GAD methods have also been proposed. Most of these methods focus on the design of a new graph neural network (Dou et al., 2020; Liu et al., 2021a; Tang et al., 2022; Wang et al., 2023; Zhuo et al., 2024) or propose a reasonable self-training mechanism for further representation learning (Wang et al., 2021; Chen et al., 2024b), followed by a binary classifier.

Although recent GAD methods have shown promising results, the inherent diversity in graph datasets is still under-explored and leads to the limitation of existing methods. The performance of unsupervised methods is limited when handling the complex patterns in real-world graphs. On the other hand, existing supervised methods are all constructed as binary classifiers, with more fine-grained subclasses in both normal and abnormal nodes unexplored. Specifically, anomaly nodes can be divided into contextual and structural nodes according to the cause of the irregularity, where contextual anomalies refer to nodes whose attributes are vastly different from those of regular nodes and structural anomalies refer to nodes with different connectivity patterns compared to other nodes(Roy et al., 2024).

Furthermore, both normal nodes and abnormal nodes can be partitioned into more fine-grained subcategories in real-world scenarios. Fig.1(a) shows an example of the review network in an E-commerce platform, where abnormal reviews can be malicious negative, fake positive, edited via a template and so on. Similarly, normal reviews can be genuinely positive, genuinely negative, neutral and so on. With such fine-grained subcategories, modeling the anomaly detection as a binary classification task will be problematic, as shown in Fig.1(b), especially when the labeled data cover

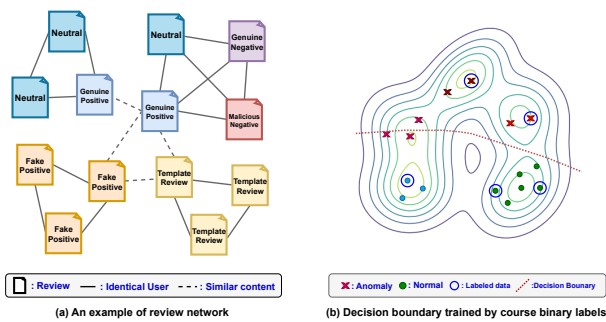

Figure 1: Node diversity and its effect in graph anomaly detection

a small ratio. With the guidance of coarse and incomplete labeled data, the decision boundary tends to deviate from the optimal one, neglecting fine-grained subcategories.

To address the challenges, we propose a novel GAD framework to capture node diversity with only a few labeled data. Based on an anomaly detector trained with limited supervision, we explore the diversity across the whole graph with two key modules. For the anomaly diversity brought by contextual and structural irregularity, we design a unified gating module to adaptively adjust the weight of node-centric attributes and neighborhood information. Considering that the existence of normal and abnormal nodes is highly related to the graph spectrum (Tang et al., 2022; Xu et al., 2024), we extract the frequency-aware information for the gating module. To capture fine-grained subcategories in both normal and abnormal nodes, we incorporate an efficient clustering module applied to node representations. Considering that the binary labels are precise but coarse while the clustering results are less accurate but offer finer-grained distinctions, we leverage the strengths of the two modules by maintaining two high-confidence sets for normal and abnormal nodes, respectively. Two sets are initialized with the labeled data and updated according to both the anomaly score function and the clustering results. The two sets will benefit the anomaly detector and the clustering process in turn. Extensive experiments on seven public datasets demonstrate the effectiveness of our framework.

The contributions of this paper can be summarized as follows:

- Considering that the contextual and structural anomalies rely on different information (e.g., centric node vs neighborhood) for detection, we propose a **unified gating classifier** to adaptively balance node-centric attributes and neighborhood signals via a frequency-aware gating mechanism, allowing a single framework to identify various anomalous patterns.
- To further take the **node sub-categories** into account, a **classifier-clustering synergy** framework is developed. By introducing two dynamic high-confidence sets initialized with the small labeled data, we use them to guide an adaptive clustering module to discover subcategories, which in turn are used to iteratively expand the sets and provide stronger, cluster-aware pseudo-label supervision to reinforce the classifier.
- Extensive experiments on seven public datasets with limited labeled data show that our method consistently outperforms recent state-of-the-art approaches, demonstrating its effectiveness in real-world GAD scenarios.

## 2 RELATED WORK

### 2.1 GRAPH ANOMALY DETECTION

Graph anomaly detection has attracted substantial research due to its importance in real-life applications. With the development of graph neural networks (GNNs) in graph mining (Kipf & Welling, 2016; Veličković et al., 2017; Hamilton et al., 2017), numerous GNN-based methods for graph anomaly detection have emerged. These methods can be categorized based on their supervision settings. The first category operates under unsupervised learning, where no labeled data are available,

and reconstruction methods dominate most of the research(Ding et al., 2019; Fan et al., 2020; Peng et al., 2020; Yuan et al., 2021; He et al., 2024; Roy et al., 2024). DOMINANT(Ding et al., 2019) uses a GCN-based encoder to learn hidden node embeddings and then reconstructs both the original graph structure and node attributes, with reconstruction loss serving as the anomaly score. AnomalyDAE (Fan et al., 2020) and GUIDE (Yuan et al., 2021) decouple the encoding of node attributes and structure. GAD-DR (Roy et al., 2024) extends reconstruction by incorporating neighborhood distribution reconstruction. ADA-GAD (He et al., 2024) generates graphs with lower anomaly rates to benefit the training performance and improve the reconstruction mechanism. Apart from reconstruction-based approaches, graph contrastive learning (Liu et al., 2021b; Chen et al., 2024a), one-class (Qiao & Pang, 2024), and community-adhere methods (Chakrabarti, 2004; Leung & Leckie, 2005) are also prevalent and show promising results.

In practical scenarios, partially labeled graphs might also be available, where labels provide more accurate insight into anomalies in the target graph. Numerous studies have explored methods under supervised settings. SemiGNN (Wang et al., 2019) incorporates a training mechanism similar to GraphSAGE (Hamilton et al., 2017) for labeled and unlabeled data. DCI (Wang et al., 2021) proposed a graph contrastive learning to handle the unlabeled data. CARE-GNN (Dou et al., 2020) and GHRN (Gao et al., 2023) adjust the heterophilic edges in the original graphs to enhance the training process. BWGNN (Tang et al., 2022) and SEC-GFD (Xu et al., 2024) incorporate spectral-based approaches with multi-band filters. GAGA (Wang et al., 2023) and PMP (Zhuo et al., 2024) design specific aggregation modules tailored for labeled normal samples, anomalous samples, and unlabeled samples within the neighborhood distribution. Despite these advances, real-world scenarios often involve only a small fraction of labeled data, posing challenges for methods requiring extensive supervision. To address this, methods like ConsisGAD (Chen et al., 2024b) incorporate consistency training for unlabeled data to mitigate the limitations of sparse labels. GGAD (Qiao et al., 2024) and CGENGA (Ma et al., 2024) leverage data-centric modules to generate auxiliary nodes, enhancing training. However, limited attention has been given to addressing the diversity in graph datasets under weak supervision.

## 2.2 LARGE-SCALE GRAPH CLUSTERING

Graph clustering aims to produce a partition where nodes in the same group are similar while those in distinct groups exhibit dissimilarity. As a fundamental task in machine learning, clustering helps uncover high-level semantics in an unsupervised learning setting. Most graph clustering methods follows a paradigm: node embeddings are learned through graph representation learning methods and a certain clustering module is applied. However, these approaches struggle with scalability on large graphs (Tu et al., 2021; Gong et al., 2022; Ding et al., 2023). To address scalability issues, $S^3GC$ (Devvrit et al., 2022) introduces a graph contrastive learning approach combined with a random walk sampler and applies K-means clustering on the learned representations. To mitigate the suboptimal solutions inherent in two-stage methods, Dink-net (Liu et al., 2023) proposes an end-to-end framework for large-scale graphs inspired by the theory of universe expansion. MAGI (Liu et al., 2024) bridges graph contrastive learning and modularity maximization to achieve scalable and efficient graph clustering.

## 3 PROBLEM DEFINITION

An attributed graph can be represented as $\mathcal{G} = \{\mathcal{V}, \{E_t\}_{t=1}^T, X, \mathcal{Y}\}$. Here, $\mathcal{V} = \{v_i\}_{i=1}^N$ denotes the set of nodes with $N$ total nodes. $X \in \mathbb{R}^{N \times d}$ represents the feature matrix of nodes, where $d$ indicates the dimension and the $i$-th row $x_i \in \mathbb{R}^d$ is the attributes of the $i$-th node. $E_t$ represents the edge set under the $t$-th relation and $T$ indicates the number of relations. For each relation, the edge set can always be represented by an adjacency matrix $A \in \{0, 1\}^{N \times N}$, where the element $A_{ij} = 1$ when there is an edge between the $i$-th and $j$-th nodes, otherwise $A_{ij} = 0$. Let $D$ denote the diagonal degree matrix where $D_{ii} = \sum_j A_{ij}$. The graph Laplacian is defined as $L = D - A$. $\mathcal{Y} = \{y_i \mid i \in \mathcal{V}\}$ is divided into a small subset $\mathcal{Y}_L$ which can be leveraged and a unavailable subset $\mathcal{Y}_U$, where $|\mathcal{Y}_L| \ll |\mathcal{Y}_U|$. In $\mathcal{Y}_L$, $y_i = 1$ indicates the $i$-th node is abnormal and $y_i = 0$ means a normal node. Our objective is to train a model $\mathcal{F}$ to detect unknown anomalous nodes in the graph.

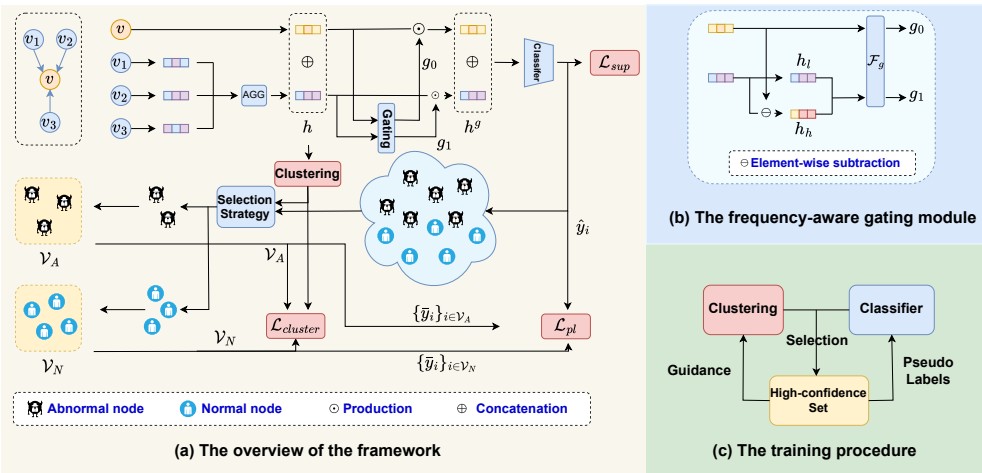

Figure 2: **(a)** The overall architecture is comprised of two key modules: a frequency-aware gating module that adaptively adjusts the weights of different information sources, and a clustering module that captures fine-grained sub-classes within the graph. These two modules are interplayed via two high-confidence node sets. **(b)** The frequency-aware gating module leverages both low-frequency and high-frequency signals to generate source-specific weights. **(c)** The two high-confidence node sets serve as a bridge that connects the classification and clustering modules.

## 4 METHOD

In this section, we provide a comprehensive explanation of the proposed framework. To offer a high-level understanding of our model, we illustrate the pipeline of the whole framework in Fig.2, which contains two main modules to capture the node diversity from distinct aspects. The frequency-aware gating module attempts to highlight the anonalous characteristics of contextual and structural anomalies by using a gate to adaptively adjust the weights of features from self-node and neighboring nodes. On the other side, a clustering module is incorporated to automatically discover fine-grained subcategories and mutually reinforce the anomaly classifier. Next, we will provide a detailed explanation of all modules.

### 4.1 A UNIFIED GATING CLASSIFIER FOR DIVERSE TYPES OF ANOMALIES

Given a graph $\mathcal{G} = \left\{ V, \{E_t\}_{t=1}^T, X \right\}$ with $T$ types of relation, we employ $(T+1)$ MLPs to project node features $X \in \mathbb{R}^{N \times d_h}$ to node-centric representation $H^c \in \mathbb{R}^{N \times d_h}$ and relation-relevant representations $\{H^{(t)} \in \mathbb{R}^{N \times d_h}\}_{t=1}^T$, with their $i$-th rows $h_i^c$ and $h_i^{(t)}$ denoting the corresponding representations of node $v_i$. To provide the model with direct and uncorrupted access to the distinct signals for both contextual and structural anomalies, we concatenate the node-centric representation and the neighborhood representation together rather than simply summing or averaging them as done in GCN. Specifically, for the node $v_i$, we compute its neighborhood representation under the $t$-th relation as $z_i^t = \sum_{j \in \mathcal{N}_i^{(t)}} h_j^{(t)}$, where $\mathcal{N}_i^{(t)}$ denotes the set of neighbors of node $v_i$ under the $t$-th relation. Once obtaining $h_i^c$ and $\{z_i^t\}_{t=1}^T$, we concatenate them to yield the node representation as

$$h_i = [h_i^c, ||_{t=1}^T z_i^t], \tag{1}$$

where $||$ means the concatenation along the indices, and $h_i \in \mathbb{R}^{d_h(1+T)}$.

By noting that contextual and structural anomalies often exhibit significantly different characteristics on their attributes and neighborhood, it is important to automatically highlight the information that is most relevant to anomalies in the representation $h_i$. To this end, we propose to learn a gate vector $g^{(i)} \in \mathbb{R}^{T+1}$ (Dauphin et al., 2017; Ma et al., 2018) that acts as a soft selection mechanism, assigning a dynamic weight to the node-centric representation $h_i^c$ and each of the aggregated neighborhood representations $z_i^t$, with the specific method to learn the vector elaborated at the end of this section.

After obtaining the gate vector $g^{(i)}$, we use it to adaptively select the representations that are most relevant to the anomaly signal, giving rise to the following anomaly-relevant representation

$$h_i^g = [g_0^i h_i^c, ||_{t=1}^T g_t^i z_i^t]. \tag{2}$$

where $g_j^{(i)} \in (0, 1)$ is a scalar, and is the $j$-th element in $g^{(i)}$. For nodes with labels, we train a binary classifier $\mathcal{F}_{cls}$ on them to detect anomalies as $\hat{y}_i = \mathcal{F}_{cls}(h_i^g)$, where $\hat{y} \in [0, 1]$ is the anomaly score. Considering the class imbalance in most GAD datasets(Chen et al., 2024b), we adopt the focal loss(Ross & Dollár, 2017) to adaptively focus on the hard samples:

$$\mathcal{L}_{sup}^{(i)} = -[(1 - \hat{y}_i)^\gamma y_i \log \hat{y}_i + \hat{y}_i^\gamma (1 - y_i) \log(1 - \hat{y}_i)], \tag{3}$$

where $\gamma$ is the hyper-parameter to adjust the contribution of each sample.

To ensure the gate vector $g^{(i)}$ effectively highlights anomalous signals, motivated by the phenomenon that graph anomalies are highly related to the graph frequency (Tang et al., 2022), its computation is specifically designed to be sensitive to information in the graph's frequency domain. Taking the $t$-th type of edges as an example, we compute the high-frequency and low-frequency components using two designed filters (Luan et al., 2022). Specifically, we adopt the random walk normalized Laplacian $L^{rw} = D^{-1}L = I - D^{-1}A$ to extract high-frequency features, while using the affinity matrix $A^{rw} = I - L^{rw} = D^{-1}A$ to extract the low-frequency component, as shown below

$$h_{i,l}^{(t)} = A_i^{rw} H^{(t)} = \frac{1}{|\mathcal{N}_i^{(t)}|} \sum_{j \in \mathcal{N}_i^{(t)}} h_j^{(t)}, \quad h_{i,h}^{(t)} = L_i^{rw} H^{(t)} = h_i^{(t)} - \frac{1}{|\mathcal{N}_i^{(t)}|} \sum_{j \in \mathcal{N}_i^{(t)}} h_j^{(t)}, \tag{4}$$

where $h_{i,l}^{(t)}$ and $h_{i,h}^{(t)}$ denote the low and high-requency representations extracted under the $t$-th relation. Next, $h_{i,l}^{(t)}$ and $h_{i,h}^{(t)}$ are concatenated and fed into an MLP-based projector $\mathcal{F}_m$ to obtain $h_{i,m}^{(t)}$. We then employ a linear layer, in conjunction with a sigmoid function, to compute the weights of the representations from the center node and its neighbors (Dauphin et al., 2017; Ma et al., 2018):

$$g^{(i)} = \sigma([h_i^c, ||_{t=1}^T h_{i,m}^{(t)}]W_g + b_g), \tag{5}$$

where $W_g \in \mathbb{R}^{(1+T)d_h \times (1+T)}$, $b_g \in \mathbb{R}^{(1+T)}$ and $\sigma(\cdot)$ is the sigmoid function.

## 4.2 Classifier-Clustering Synergy for Node Sub-categories Discovery

Although the gating module helps to capture different kinds of anomalies, the binary classification task still overlooks the inherent diversity in graph data. Both normal and abnormal nodes contain finer-grained subcategories. Under coarse binary supervision, the classifier may learn a sub-optimal decision boundary, leading to under-exploration and misclassification of certain subclasses. The small ratio of labeled data will further make the problem worse.

To capture the inherent node diversity present in the graph data, we introduce a framework built on a core principle of synergy between the binary classifier and the unsupervised clustering module. Compared to the binary classifier trained with precise yet coarse-grained labels, the clustering module provides less accurate but more fine-grained structural insights. This motivates us to leverage the complementary strengths of both the clustering results and the classifier. We achieve this through two dynamically maintained node sets: $\mathcal{V}_A$ and $\mathcal{V}_N$, representing high-confidence abnormal and normal nodes, respectively. Based on these sets, we design a training procedure as shown in Fig.2(c), these sets are initialized with labeled data and then iteratively refined through a process of mutual reinforcement: the classifier's predictions guide the clustering, and the resulting cluster structures enhance the classifier. More details are presented below.

**Clustering Module** Given two sets of high-confidence nodes $\mathcal{V}_A^e$ and $\mathcal{V}_N^e$ at the $e$-th training epoch, we leverage them to guide the clustering process. Recognizing that the number of fine-grained clusters is unknown and highly dataset-dependent, as confirmed by our ablation studies (see Table 6), we propose a method that can adaptively discover the anomalous and normal clusters, as well as their numbers $K_a^e$ and $K_n^e$, at a given epoch $e$. First, two sets of learnable cluster centers $\{C_i^{e,a} \in \mathbb{R}^{(1+T)d_h}\}_{i=1}^{i=K_a^e}$ and $\{C_i^{e,n} \in \mathbb{R}^{(1+T)d_h}\}_{i=1}^{i=K_n^e}$ are updated through the application of DBSCAN (Ester et al., 1996) on node representations $\{h_i \mid v_i \in \mathcal{V}_A^e\}$ and $\{h_i \mid v_i \in \mathcal{V}_N^e\}$,

respectively. To encourage separability, we maximize the distance between the two sets of cluster centers through a dilation loss:

$$\mathcal{L}_{dilation} = -\frac{1}{K_n^e K_a^e} \sum_{i=1}^{K_n^e} \sum_{j=1}^{K_a^e} ||C_i^{e,n} - C_j^{e,a}||_2^2. \tag{6}$$

Next, we promote compactness by encouraging node representations to be close to the cluster centers. Since the nodes in $\mathcal{V}_N^e$ and $\mathcal{V}_A^e$ are believed to be more reliable and discriminative, we push the representations of these nodes to their nearest cluster centers. For the remaining low-confidence nodes, we compute their average distance to both sets of cluster centers and use the smaller one. This leads to the shrinking loss on a mini-batch $B$:

$$\mathcal{L}_{shrink}^B = \frac{1}{|B^N|} \sum_{i \in B^N} \min_{j \in \{1,2,...,K_n^e\}} ||h_i - C_j^{e,n}||_2^2 + \frac{1}{|B^A|} \sum_{i \in B^A} \min_{j \in \{1,2,...,K_a^e\}} ||h_i - C_j^{e,a}||_2^2$$
$$+ \frac{1}{|B^{LC}|} \sum_{i \in B^{LC}} \min(\frac{1}{K_n^e} \sum_{j=1}^{K_n^e} ||h_i - C_j^{e,n}||_2^2, \frac{1}{K_a^e} \sum_{j=1}^{K_a^e} ||h_i - C_j^{e,a}||_2^2), \tag{7}$$

where $B^N$, $B^A$, and $B^{LC}$ represent the subsets of $B$ corresponding to $\mathcal{V}_N$, $\mathcal{V}_A$, and the remaining low-confidence nodes, respectively. Following prior work (Liu et al., 2023; Nickerson, 1998), we avoid forcing all low-confidence nodes toward cluster centers to mitigate confirmation bias. The complete cluster-aware loss is:

$$\mathcal{L}_{cluster}^B = \mathcal{L}_{dilation} + \mathcal{L}_{shrink}^B. \tag{8}$$

**Pseudo-label Mechanism** To further assist the anomaly detector and gating module, we generate pseudo-labels from high-confidence nodes. Specifically, for node $v_i$, we assign $\bar{y}_i = 1$ if $v_i \in \mathcal{V}_A$ and $\bar{y}_i = 0$ if $v_i \in \mathcal{V}_N$. The pseudo-label loss is formulated as:

$$\mathcal{L}_{PL} = \sum_{i \in \mathcal{V}_A \cup \mathcal{V}_N} -[(1-\hat{y}_i)^\gamma \bar{y}_i \log \hat{y}_i + \hat{y}_i^\gamma (1-\bar{y}_i) \log(1-\hat{y}_i)], \tag{9}$$

where $\gamma$ is consistent with Eq. (3). By incorporating reliable pseudo-labeled data, the anomaly detector equipped with the gating module can be better trained with richer supervision.

**Updating of High-Confidence Sets** To expand the high-confidence sets with more accurate samples, an alternative strategy is to leverage both the clustering module and the classifier. In practice, we employ the classifier to select top and bottom-scoring samples as candidate nodes. Next, we calculate the Euclidean distances between each candidate node and the cluster centers $\{C_k^{n,e}\}_{k=1}^{K_n^e}$ and $\{C_k^{a,e}\}_{k=1}^{K_a^e}$. A candidate node $v_i$ is added to the high-confidence abnormal set $\mathcal{V}_A$ if it satisfies the following condition:

$$(s_i > s_p) \wedge (\min_j ||h_i - C_j^{e,n}||_2^2 > \min_j ||h_i - C_j^{e,a}||_2^2). \tag{10}$$

where $s_i$ is node $v_i$'s anomaly scores, and $s_p$ is the threshold for high-confidence scores. The high-confidence normal set $\mathcal{V}_N$ is expanded in a similar manner. By integrating both the clustering module and classifier, this strategy ensures that the expansion of high-confidence node sets leverages information from both labeled data and cluster-aware diversity, leading to a more effective selection process.

### 4.3 TRAINING PROCEDURE

It can be observed that our proposed framework incorporates both supervised loss on a few labeled data and unsupervised loss on the entire dataset. To make the training process scalable across datasets of varying sizes, we adopt the approach from ConsisGAD (Chen et al., 2024b), training the module on two batches in each iteration: one sampled from the labeled data and the other from the whole dataset. We refer to these batches as $B_S$ and $B_U$, respectively. The overall loss function for one batch can be expressed as

$$\mathcal{L}^{B_S+B_U} = \mathcal{L}_{sup}^{B_S} + \lambda_1 \mathcal{L}_{PL}^{B_U} + \lambda_2 \mathcal{L}_{cluster}^{B_U}, \tag{11}$$

Table 1: Dataset Statistics

| Dataset | # Nodes | # Edges | Anomaly (%) | # Features |
|---------|---------|---------|-------------|------------|
| Weibo | 8,405 | 407,943 | 10.3% | 400 |
| Tolokers | 11,758 | 519,000 | 21.8% | 10 |
| Elliptic | 203,769 | 234,355 | 4.6% | 166 |
| Amazon | 11,944 | 4,398,392 | 6.87% | 25 |
| YelpChi | 45,954 | 3,846,979 | 14.35% | 32 |
| T-Finance | 39,357 | 21,222,543 | 4.58% | 10 |
| T-Social | 5,781,065 | 73,105,508 | 3.01% | 10 |

Table 2: Performance (%) comparison on Weibo, Tolokers, and Elliptic datasets.

| Methods | Weibo (1%) | | | Tolokers (1%) | | | Elliptic (1%) | | |
|---------|------------|--------|---------|---------------|--------|---------|---------------|--------|---------|
| | AUROC | AUPRC | Macro F1 | AUROC | AUPRC | Macro F1 | AUROC | AUPRC | Macro F1 |
| MLP | 62.78±9.73 | 48.65±2.67 | 72.28±0.23 | 70.32±0.73 | 35.09±0.55 | 58.74±1.16 | 88.59±0.09 | 24.11±1.28 | 68.46±1.34 |
| GCN | 73.08±10.4 | 63.6±3.53 | 82.18±1.81 | 67.03±2.28 | 34.28±2.44 | 57.6±1.91 | 70.37±2.34 | 4.86±0.80 | 50.82±1.76 |
| GAT | 67.32±1.54 | 27.39±4.01 | 62.92±4.66 | 68.8±0.47 | 33.52±1.24 | 58.47±0.78 | 67.23±3.40 | 4.20±1.02 | 47.97±5.88 |
| GraphSAGE | 52.82±7.58 | 37.55±6.36 | 68.52±2.53 | 71.55±0.79 | 36.2±0.93 | 61.35±0.55 | 84.81±3.27 | 18.66±7.83 | 62.92±5.48 |
| GIN | 70.77±16.27 | 37.33±16.83 | 63.69±5.29 | 56.51±7.43 | 25.77±4.15 | 52.45±5.97 | 66.11±5.98 | 5.08±1.45 | 50.39±1.82 |
| CARE-GNN | 81.49±1.56 | 47.84±1.44 | 54.96±2.01 | 72.56±0.57 | 36.23±0.48 | 47.09±0.59 | 80.39±5.56 | 16.94±4.24 | 30.38±6.53 |
| PC-GNN | 82.23±0.62 | 54.23±3.42 | 74.24±2.17 | 68.19±0.78 | 32.56±0.93 | 38.5±18.95 | 81.16±1.53 | 15.27±1.73 | 53.47±4.14 |
| BWGNN | 82.65±4.14 | 49.74±2.76 | 81.9±1.42 | 61.09±0.61 | 28.00±0.42 | 59.95±0.48 | 76.13±0.64 | 23.63±0.68 | 72.88±0.41 |
| GAGA | 87.8±1.19 | 58.13±3.84 | 53.15±12.14 | 67.3±5.49 | 31.99±3.76 | 50.00±5.53 | 83.45±2.6 | 22.14±4.78 | 70.72±7.88 |
| ConsisGAD(GNN) | 91.47±0.7 | 68.75±3.23 | 77.68±0.39 | 70.48±0.64 | 35.98±0.67 | 60.29±0.2 | 88.11±0.96 | 34.98±0.94 | 70.03±1.58 |
| ConsisGAD | 85.99±2.88 | 64.26±3.13 | 77.46±1.33 | 71.61±0.48 | 37.52±0.75 | 60.79±0.48 | 89.99±0.38 | 37.72±1 | 72.1±1.19 |
| PMP | 73.65±1.17 | 66.92±1.00 | 42.09±2.14 | 69.91±1.61 | 38.23±1.67 | 53.05±6.09 | 77.56±1.67 | 14.62±1.78 | 63.16±1.59 |
| LEX-GNN | 92.26±1.37 | 74.15±2.19 | 77.47±4.5 | 71.31±1.12 | 36.13±1.62 | 43.88±0.13 | 88.11±1.00 | 24.11±2.51 | 65.69±0.57 |
| Ours | **94.40±0.40** | **81.30±2.58** | **85.66±1.80** | **73.15±1.08** | **40.57±1.37** | **62.17±0.98** | **90.76±0.31** | **42.03±0.97** | **74.34±0.33** |

Table 3: Performance (%) comparison on Amazon, YelpChi, T-Finance and T-social datasets.

| Methods | Amazon (1%) | | | YelpChi (1%) | | | T-Finance (1%) | | | T-social (0.01%) | | |
|---------|-------------|--------|---------|--------------|--------|---------|----------------|--------|---------|-------------------|--------|---------|
| | AUROC | AUPRC | Macro F1 | AUROC | AUPRC | Macro F1 | AUROC | AUPRC | Macro F1 | AUROC | AUPRC | Macro F1 |
| MLP | 92.39±0.72 | 79.37±1.83 | 87.53±1.61 | 72.18±0.39 | 31.09±0.52 | 61.61±0.33 | 92.17±0.64 | 52.79±5.41 | 82.33±0.54 | 66.95±0.71 | 6.00±0.33 | 54.09±0.61 |
| GCN | 87.34±0.59 | 48.06±2.73 | 70.94±2.43 | 54.65±0.53 | 17.07±0.44 | 35.59±10.27 | 89.29±0.19 | 53.94±3.22 | 77.16±1.20 | 83.30±1.60 | 23.79±2.43 | 65.16±0.92 |
| GAT | 80.74±3.64 | 45.46±11.09 | 63.45±12.82 | 70.14±1.91 | 28.90±1.98 | 61.22±1.32 | 87.40±4.41 | 75.49±5.63 | 63.45±12.82 | 73.46±3.32 | 13.47±2.83 | 61.98±2.06 |
| GraphSAGE | 90.12±0.48 | 73.17±4.65 | 84.25±2.26 | 73.70±0.52 | 34.57±0.78 | 63.33±0.51 | 89.42±1.36 | 49.08±6.34 | 77.62±1.87 | 71.45±2.24 | 8.73±0.91 | 56.47±0.64 |
| GIN | 84.35±0.75 | 39.96±2.00 | 71.20±1.37 | 56.98±0.82 | 18.34±0.64 | 53.58±0.41 | 81.29±1.66 | 21.66±3.98 | 65.38±3.05 | 78.70±2.19 | 16.24±5.53 | 61.62±5.93 |
| CARE-GNN | 89.68±0.76 | 50.56±3.96 | 75.74±0.50 | 72.11±1.23 | 31.09±1.71 | 61.62±0.87 | 91.45±0.40 | 72.27±1.09 | 83.68±0.78 | -OOM- | -OOM- | -OOM- |
| PC-GNN | 91.18±0.66 | 77.92±1.49 | 85.25±2.09 | 75.17±0.44 | 36.60±0.91 | 64.23±0.47 | 91.74±0.85 | 74.77±0.98 | 86.97±0.24 | 64.68±0.64 | 4.30±0.09 | 49.66±0.12 |
| BWGNN | 88.56±0.87 | 79.26±1.11 | **90.48±0.98** | 77.62±2.37 | 39.87±1.79 | 66.54±0.73 | 93.08±1.57 | 77.79±3.87 | 86.97±1.51 | 84.40±3.01 | 49.96±3.75 | 76.37±1.82 |
| GAGA | 82.61±6.87 | 56.59±6.60 | 76.85±8.08 | 71.61±2.13 | 31.96±3.37 | 61.81±1.69 | 92.36±1.45 | 64.34±6.01 | 81.10±2.60 | 78.92±1.26 | 23.72±4.81 | 65.58±3.30 |
| ConsisGAD(GNN) | 92.01±0.71 | 78.49±0.40 | 85.53±0.51 | 80.95±0.36 | 43.25±0.31 | 67.62±0.31 | 94.72±0.11 | 83.92±0.15 | 89.73±0.38 | 93.54±0.35 | 53.40±1.28 | 76.45±1.06 |
| ConsisGAD | 93.91±0.58 | 83.33±0.34 | 90.03±0.53 | 83.36±0.53 | 47.33±0.58 | 69.72±0.30 | 95.33±0.30 | 86.63±0.44 | 90.97±0.63 | 94.31±0.20 | 58.38±2.10 | 78.08±0.54 |
| PMP | 91.82±0.74 | 66.92±1.00 | 87.72±1.15 | 80.03±2.07 | 44.36±1.91 | 66.93±1.4 | 94.11±0.4 | 80.31±1.43 | 87.83±0.41 | 93.75±1.01 | 51.35±1.17 | 78.15±2.04 |
| LEX-GNN | 93.02±0.21 | 82.12±0.41 | 87.33±1.76 | 83.14±0.53 | 39.96±0.68 | 69.73±0.68 | 92.76±1.19 | 62.49±7.93 | 55.76±13.86 | 80.39±2.44 | 13.73±2.64 | 58.65±4.15 |
| Ours | **94.99±0.47** | **84.62±0.69** | 89.37±0.66 | **83.7±0.61** | **48.27±0.30** | **70.18±0.13** | **96.43±0.12** | **87.60±0.53** | **91.42±0.38** | **95.09±0.12** | **60.24±0.03** | **79.23±0.34** |

where $\lambda_1$ and $\lambda_2$ are hyperparameters that control the weight of each component. During training, we update $\mathcal{V}_A$ and $\mathcal{V}_N$ from the unlabeled data according to the anomaly scores generated by the detector. The nodes in $\mathcal{V}_N$ and $\mathcal{V}_A$ can be leveraged to update the cluster centers for every certain epochs. To better clarify the training procedure of our framework, a pseudo-code is provided in the Appendix A.

## 5 EXPERIMENTS

### 5.1 EXPERIMENTS SETTINGS

We evaluate our method on seven public real-world datasets against classic GNNs and recent state-of-the-art GAD methods. The statistics of the datasets are summarized in Table 1. To evaluate our method under limited supervision, we follow the settings in Chen et al. (2024b) and use a low training ratio across all datasets. **Detailed descriptions of the datasets, baselines, and our full implementation and evaluation details are provided in Appendix B.**

### 5.2 OVERALL COMPARISON

We summarize all experimental results in Tables 2 and 3. The best performance on each dataset under each metric is highlighted in bold. "OOM" indicates that the model ran out of memory during training. As shown, our method achieves the best performance in the majority of cases. For example,

Table 4: Ablation studies on Weibo, Tolokers, Elliptic datasets.

| Module | | | Weibo (1%) | | | Tolokers (1%) | | | Elliptic (1%) | | |
|---|---|---|---|---|---|---|---|---|---|---|---|
| Gating | Clustering | Pseudo Label | AUROC | AUPRC | Macro F1 | AUROC | AUPRC | Macro F1 | AUROC | AUPRC | Macro F1 |
| ✓ | | | 90.83±1.28 | 70.51±2.94 | 80.62±1.56 | 71.62±0.27 | 37.32±1.24 | 61.22±0.87 | 90.05±0.74 | 37.86±1.29 | 72.89±0.60 |
| ✓ | ✓ | | 93.34±0.65 | 74.97±0.72 | 82.93±1.00 | 71.64±1.10 | 39.85±1.00 | 61.36±0.96 | 90.36±0.26 | 40.73±1.48 | 73.47±0.52 |
| | ✓ | ✓ | 92.86±1.19 | 77.86±3.00 | 84.45±1.16 | 71.90±1.24 | 39.83±1.43 | 61.26±1.29 | 89.90±0.30 | 40.00±1.27 | 74.11±0.31 |
| ✓ | ✓ | ✓ | 94.40±0.40 | 81.30±2.58 | 85.66±1.80 | 73.15±1.08 | 40.57±1.37 | 62.17±0.98 | 90.76±0.31 | 42.03±0.97 | 74.34±0.33 |

Table 5: Ablation studies on Amazon, Yelp, T-Finance datasets.

| Module | | | Amazon (1%) | | | Yelp (1%) | | | T-Finance (1%) | | |
|---|---|---|---|---|---|---|---|---|---|---|---|
| Gating | Clustering | Pseudo Label | AUROC | AUPRC | Macro F1 | AUROC | AUPRC | Macro F1 | AUROC | AUPRC | Macro F1 |
| ✓ | | | 91.00±0.66 | 79.71±0.93 | 87.41±0.27 | 82.77±0.40 | 44.91±1.04 | 68.35±0.61 | 95.64±0.64 | 84.90±1.19 | 90.19±0.61 |
| ✓ | ✓ | | 94.62±0.76 | 83.55±1.13 | 88.53±1.17 | 83.31±0.36 | 46.77±1.01 | 69.25±0.23 | 96.40±0.24 | 86.80±0.69 | 90.45±0.60 |
| | ✓ | ✓ | 95.67±0.19 | 82.32±1.18 | 86.93±1.25 | 83.56±0.27 | 46.13±0.63 | 69.10±0.29 | 96.40±0.23 | 85.94±0.91 | 89.80±0.58 |
| ✓ | ✓ | ✓ | 94.99±0.47 | 84.62±0.69 | 89.37±0.66 | 83.70±0.61 | 48.27±0.30 | 70.18±0.13 | 96.43±0.12 | 87.60±0.53 | 91.43±0.38 |

on the Weibo dataset, our method outperforms the second-best method by relative margins of 2.32%, 9.64%, and 4.59% in AUROC, AUPRC, and Macro-F1, respectively. On the Tolokers dataset, our method achieves relative improvements of 0.8%, 6.1%, and 2.27% in the same metrics. Similarly, on the Elliptic dataset, it shows relative gains of 0.86%, 11.43%, and 2.00%. Significant improvements are also observed on other datasets, which collectively validate the effectiveness of our approach. It is worth noting that among classical models, GNNs do not consistently outperform traditional MLPs. Our method integrates node and neighborhood representations via concatenation and utilizes a frequency-aware gating module to adaptively balance different information sources, resulting in superior performance. Among recent graph anomaly detection methods, ConsisGAD is tailored for weakly supervised settings and performs competitively in most scenarios. Other baseline methods typically assume the availability of abundant labeled data. However, in real-world applications where labeled data is limited, these methods struggle to remain competitive. In contrast, our method adaptively balances node-centric and neighborhood information via a gating mechanism and leverages clustering to identify node diversity, achieving consistently superior results across datasets.

## 5.3 ANALYSIS OF NODE DIVERSITY

To evaluate the proposed method's capability in capturing node diversity within graph data, we conduct a case study on anomalous nodes from the Weibo test set. Specifically, for each anomalous node, we concatenate its own attribute features with the average features of its neighbors to jointly represent semantic and structural information.

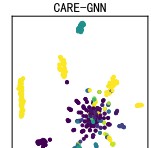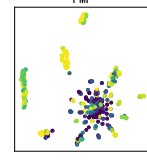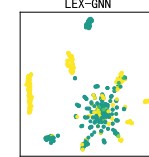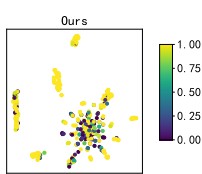

Figure 3: Visualization of anomaly detection results on diverse anomalous samples in Weibo.

These features are then projected into a 2D space using T-SNE for visualization, where each point corresponds to one test sample. We train CARE-GNN, PMP, LEX-GNN, and our method using identical data splits. Each test anomaly is assigned an anomaly score in the range $[0, 1]$. We visualize the test anomalies based on their 2D coordinates and corresponding scores in Figure 3, where brighter colors indicate higher anomaly scores. Key observations from Figure 3: (1) High diversity among anomalies: Although labeled under the same class, anomalous nodes form distinct fine-grained clusters, with high intra-cluster similarity and clear inter-cluster boundaries. (2) Superiority of our method: Compared to baselines, our method assigns more consistent and distinctive scores across nearly all anomaly clusters, effectively uncovering anomaly diversity.

## 5.4 ABLATION STUDIES

In this section, we conduct ablation studies on our framework's key components: the frequency-aware gating module, the clustering module, and the pseudo-labeling mechanism. We assess their contributions by removing each module individually. Furthermore, we investigate the framework's

Table 6: The impact of clustering methods on Amazon, T-Finance datasets.

| Method | Amazon (1%) | | | T-Finance (1%) | | |
|---|---|---|---|---|---|---|
| | AUROC | AUPRC | Macro F1 | AUROC | AUPRC | Macro F1 |
| **Kmeans(n_cluster=4)** | 93.98±0.68 | 84.20±0.27 | 88.46±0.39 | 96.94±0.31 | 86.23±0.33 | 90.12±0.61 |
| **Kmeans(n_cluster=8)** | 94.23±0.37 | 84.69±0.44 | 90.85±0.23 | 96.62±0.25 | 85.06±0.11 | 88.94±0.14 |
| **Kmeans(n_cluster=12)** | 94.60±0.27 | 84.69±0.44 | 90.85±0.23 | 96.51±0.05 | 85.05±0.97 | 89.06±0.23 |
| **Kmeans(n_cluster=16)** | 94.30±0.35 | 84.38±0.33 | 90.57±0.35 | 96.56±0.16 | 84.84±0.34 | 89.02±0.26 |
| **Ours(DBSCAN)** | 94.99±0.47 | 84.62±0.69 | 89.37±0.66 | 96.43±0.12 | 87.60±0.53 | 91.42±0.38 |

sensitivity to the clustering algorithm by replacing our default DBSCAN with K-means. Additional hyperparameter analysis is in Appendix C.

**The effect of the gating module**   As shown in the third and fourth rows of Tables 4 and 5, incorporating the gating mechanism consistently improves and balances performance across all metrics. For instance, on the Weibo dataset, it yields relative gains of 1.66%, 4.42%, and 1.43% in AUROC, AUPRC, and F1-score, respectively. Although AUROC is slightly higher without the gate on the Amazon dataset, AUPRC and Macro F1 drop notably; this indicates the gating mechanism helps the model better focus on anomalies, given that AUPRC is more sensitive to positive samples.

**The effect of clustering**   Tables 4 and 5 show that the clustering module substantially improves performance. For instance, on the Weibo dataset, it yields relative gains of 2.76%, 6.33%, and 2.87% in AUROC, AUPRC, and F1-score, respectively. This demonstrates its effectiveness in uncovering fine-grained distinctions that are difficult to capture with limited binary labels alone. The mechanism enhances the clustering structure of node representations under the guidance of high-confidence samples, which aids the classifier and supports the selection of new high-confidence samples.

**The effect of the pseudo-label mechanism**   The second and fourth rows of Tables 4 and 5 demonstrate the effectiveness of incorporating the pseudo-labeling mechanism into our framework. The pseudo-labeling mechanism consistently improves performance, particularly under weak supervision. Specifically, on the Weibo dataset, we observe relative improvements of 1.14% in AUROC, 8.44% in AUPRC, and 3.29% in F1-score. The improvements stem from the additional supervision signal provided by the pseudo-labeling strategy. By generating and iteratively refining pseudo-labels for high-confidence nodes, this mechanism alleviates the supervision bottleneck from limited ground-truth labels and promotes better feature utilization and decision boundary refinement.

**The effect of clustering methods**   To validate the robustness of our framework to the choice of clustering algorithm, we replace DBSCAN with the K-means to investigate its impact on the final anomaly detection performance. The Table 6 shows that when a suitable $K$ is selected, K-means can be an alternative for the initialization of cluster centers. For Amazon, setting $K = 12$ can achieve good results. For T-Finance, K-means with relatively lower $K$ achieve better results. However, the best choice of $K$ depends on the characteristics of specific datasets. This observation highlights the primary advantage of our default choice, DBSCAN, which obviates the need for such dataset-specific hyperparameter tuning by automatically determining the number of the clusters.

## 6   CONCLUSION

In this work, we propose a novel framework for graph anomaly detection (GAD) under weak supervision, aiming to capture node diversity in graph data and address the limitation of binary classification. The unified gating module adjusts the weights of node-centric and neighborhood representations via frequency-aware gating mechanism, capturing both contextual and structural anomalies. Additionally, a classifier-clustering synergy framework is developed, under which the discovery of node sub-categories and the classification of anomalies can mutually reinforce each other. To further enhance the learning process, we maintain two dynamic high-confidence sets for normal and abnormal nodes, which are jointly refined with the classifier and clustering module throughout training. Extensive experiments conducted on seven real-world graph datasets demonstrate the robustness and superiority of our method across multiple metrics.

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

# A    TRAINING DETAILS

## A.1    TRAINING PROCESS

Our framework is trained in a mini-batch manner to ensure scalability and leverage the Adam optimizer (Kingma & Ba, 2014). Moreover, the high-confidence node sets are updated jointly with both the classifier and the clustering module. The overall training process is detailed in Algorithm 1. Specifically, we initialize two high-confidence node sets, $\mathcal{V}_N^0$ and $\mathcal{V}_A^0$, corresponding to the labeled normal and abnormal nodes, respectively. To begin with, we warm up the GNN encoder and the classifier using only the labeled data by minimizing the supervised loss in Eq. (3) for a fixed number of epochs. This helps the encoder learn meaningful representations.

Once the encoder is sufficiently warmed up, we initialize cluster centers for both normal and abnormal nodes using the corresponding high-confidence sets. For example, to initialize the abnormal cluster centers from $\mathcal{V}_A^0$, we apply DBSCAN (Ester et al., 1996) on the representation set $\{h_i \mid i \in \mathcal{V}_A^0\}$ without requiring a pre-defined number of clusters. This yields clustering results $\{c_i \mid i \in \mathcal{V}_A^0\}$, where $c_i \in \{1, 2, \ldots, C_a^0\}$ and $C_a^0$ is the number of valid clusters detected. The mean representation of each cluster is then used as its cluster center, denoted by $\{C_i^{0,a}\}_{i=1}^{K_a^0}$. The normal cluster centers $\{C_i^{0,n}\}_{i=1}^{K_n^0}$ are computed in a similar fashion using $\mathcal{V}_N^0$.

During each training iteration, we sample a labeled batch $B^S$ from the labeled set $\mathcal{V}^S$ and an unlabeled batch $B^U$ from the entire node set $\mathcal{V}$. The supervised loss $\mathcal{L}_{sup}^{B^S}$ is computed on $B^S$ using Eq. (3). Meanwhile, the pseudo-labeling loss $\mathcal{L}_{PL}^{B^U}$ and clustering loss $\mathcal{L}_{cluster}^{B^U}$ are calculated on $B^U$ using Eq.(9) and Eq.(8), respectively. Following this, we update the high-confidence node sets based on the outputs of the classifier and clustering module using the selection strategy described in Section 4.2. Additionally, the cluster centers are periodically updated based on the latest high-confidence sets every fixed number of epochs.

## A.2    THE SETTINGS OF EACH DATASET

There are several hyperparameters in our proposed framework. Specifically, $\lambda_{pl}$ and $\lambda_{cluster}$ control the influence of the pseudo-labeling mechanism and the clustering module, respectively. Parameters $p$ and $q$ denote the threshold ratios used to select candidate nodes based on the scoring function. The parameter $\gamma$ is the focusing factor in the Focal Loss, as defined in Eq.3 and Eq.9. Training is performed in a mini-batch manner with batch size $bs$. The detailed settings of these hyperparameters for each dataset are provided in Table 7.

Table 7: Hyperparameter settings for each dataset

| Dataset | $\lambda_{pl}$ | $\lambda_{cluster}$ | $p$ | $q$ | $\gamma$ | $bs$ |
|---|---|---|---|---|---|---|
| Weibo | $1 \times 10^{-3}$ | $1 \times 10^{-1}$ | 3% | 5% | 2.0 | 128 |
| T-Finance | $1 \times 10^{-1}$ | $1 \times 10^{-3}$ | 1% | 5% | 8.0 | 128 |
| Amazon | $1 \times 10^{-3}$ | $5 \times 10^{-4}$ | 1% | 3% | 8.0 | 32 |
| YelpChi | $1 \times 10^{-1}$ | $1 \times 10^{-1}$ | 1% | 10% | 4.0 | 32 |
| Elliptic | $1 \times 10^{-3}$ | $1 \times 10^{-3}$ | 1% | 5% | 2.0 | 128 |
| Tolokers | 1.0 | $1 \times 10^{-1}$ | 1% | 5% | 1.0 | 32 |
| T-Social | $1 \times 10^{-3}$ | $1 \times 10^{-3}$ | 1% | 5% | 2.0 | 128 |

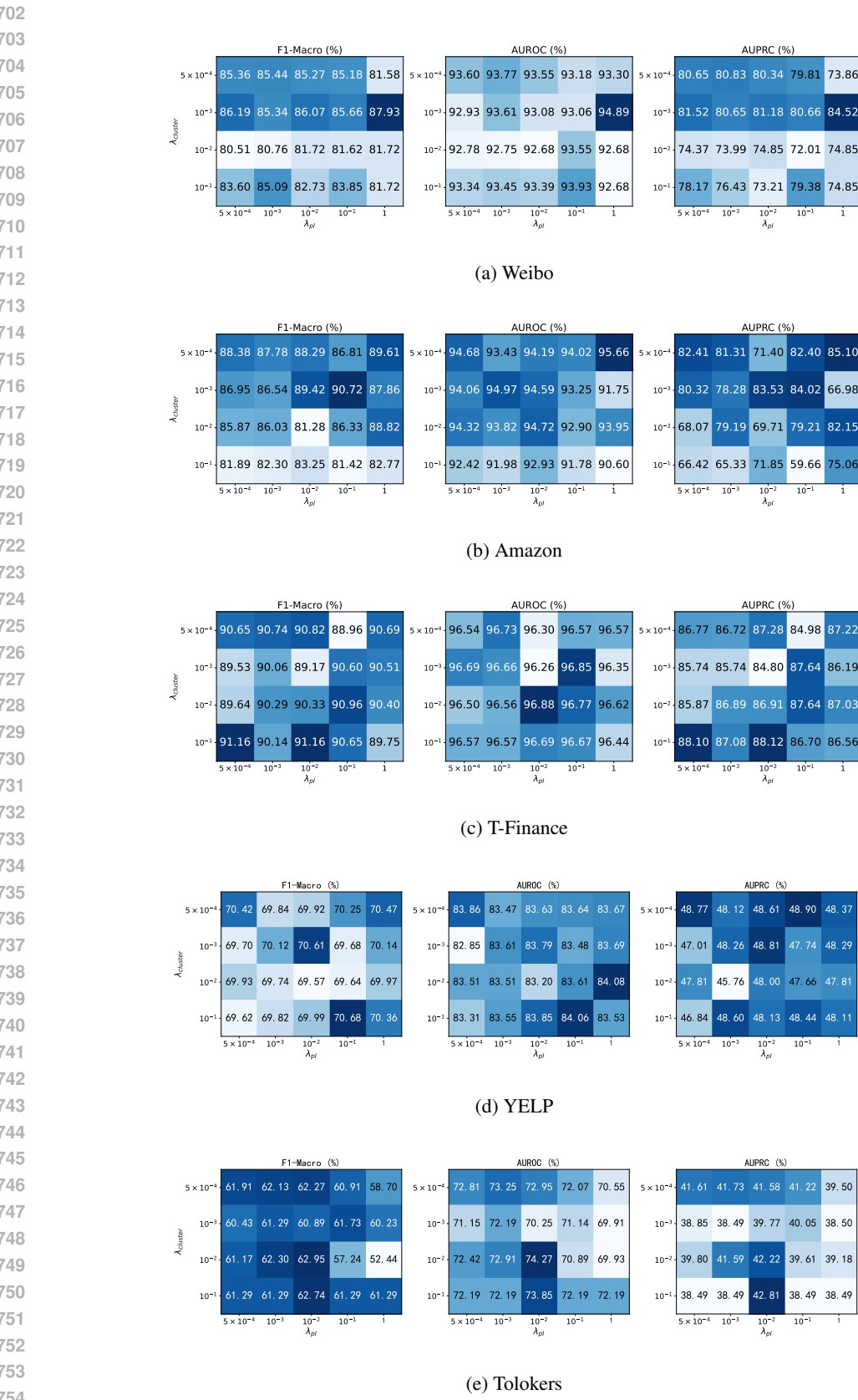

Figure 4: The performances under different setting of $\lambda_{pl}$ and $\lambda_{cluster}$.

---

**Algorithm 1** Training procedure of the framework

---

1: **Input:** $\mathcal{G} = \{\mathcal{V}, \{E_t\}_{t=1}^T, X, \mathcal{Y}^L\}$: A multi-relation graph with a small labeled set ratio, $max\_iters$: Maximum number of training iterations, $\lambda_1$: Weight of the contrastive learning loss, $\lambda_2$: Weight of the clustering loss, $D(\cdot)$: Anomaly detector with a shared feature extractor and gating mechanism, update_iters: the number of iterations to update the cluster centers explicitly, $p(q)$: the ratio of high-confidence nodes with highest (lowest) scores, $w$: the number of warm-up epochs.

2: **Output:** The anomaly score for unlabeled data.

3: Warm up the $D(\cdot)$ on labeled data $\mathcal{V}^S$ through Eq.(3) for $w$ epochs.

4: Initialize the high-confidence node set $\mathcal{V}_A = \{v_i | v_i \in \mathcal{V}^L \wedge y_i = 1\}$, $\mathcal{V}_N = \{v_i | v_i \in \mathcal{V}^L \wedge y_i = 0\}$

5: Initialize cluster centers $\{C_i^{0,a}\}_{i=1}^{K_a^0}$ and $\{C_i^{0,n}\}_{i=1}^{K_n^0}$ through DBSCAN on the node representations of $\mathcal{V}_A$ and $\mathcal{V}_N$, respectively.

6: **for** $e = 1, \ldots, max\_iters$ **do**

7:    **if** $e\%$update_iters $== 0$ **then**

8:       Update cluster centers $\{C_i^{e,a}\}_{i=1}^{K_e^a}$, $\{C_i^{e,n}\}_{i=1}^{K_e^n}$ on $\mathcal{V}_N$;

9:    **end if**

10:    Sample a batch $B^S$ from the labeled dataset $\mathcal{V}^S$;

11:    Sample a batch $B^U$ from the entire dataset $\mathcal{V}$;

12:    Compute supervised loss $\mathcal{L}_{sup}^{B^S}$ on batch $B^S$ through Eq.(3);

13:    Compute pseudo-label loss $\mathcal{L}_{PL}^{B^U}$ and clustering loss $\mathcal{L}_{cluster}^{B^U}$ on batch $B^U$ through Eq.(9)(8);

14:    Compute the anomaly scores on batch $B^U$ through the classifier, then add nodes to $\mathcal{V}_A$ if the condition 10 satisfied. The updating of $\mathcal{V}_N$ is similar.

15:    Optimize parameters in $D$ using the total loss through Eq.(11);

16: **end for**

17: $\hat{\mathcal{Y}} = D(\mathcal{G})$

18: **return** the predicted labels $\hat{\mathcal{Y}}^U$ for the unlabeled dataset $\mathcal{V}^U$.

---

# B EXPERIMENTAL SETTINGS DETAILS

## B.1 DATASET DETAILS

We evaluate our method on seven public graph anomaly detection datasets, all sourced from real-world scenarios without synthetic anomaly samples (Tang et al., 2023). The basic statistics of each dataset are summarized in Table 1 in the main text.

- **Weibo** (Zhao et al., 2020) and **T-Social** (Tang et al., 2022) represent user accounts on social media platforms.

- **Tolokers** (Platonov et al., 2023), **Amazon** (McAuley & Leskovec, 2013), and **YelpChi** (Rayana & Akoglu, 2015) are collected from crowdsourcing or E-commerce platforms. Notably, Amazon and YelpChi are heterogeneous graphs with multiple types of relations.

- **T-Finance** (Tang et al., 2022) and **Elliptic** (Weber et al., 2019) are constructed to detect fraudulent users and illicit activities in financial networks.

## B.2 BASELINE DETAILS

We compare our method with state-of-the-art approaches to demonstrate its effectiveness. The baselines are grouped into two categories:

- **Classic GNN Models**: This category includes standard models where the supervision loss is computed only on the labeled nodes. The models are: MLP (Rosenblatt, 1958), GCN (Kipf & Welling, 2016), GraphSAGE (Hamilton et al., 2017), GAT (Veličković et al., 2017), and GIN (Xu et al., 2019).

- **Recent GAD Methods**: This category consists of recent methods specifically designed for graph anomaly detection. The models include: CARE-GNN (Dou et al., 2020), PC-GNN (Liu et al.,

2021a), BWGNN (Tang et al., 2022), GAGA (Wang et al., 2023), ConsisGAD (Chen et al., 2024b), PMP (Zhuo et al., 2024), and LEX-GNN (Hyun et al., 2024). Among these, ConsisGAD is particularly relevant as it also addresses the challenge of limited supervision.

## B.3 IMPLEMENTATION AND EVALUATION DETAILS

**Implementation.** Our method is implemented in PyTorch 1.13.1 with Python 3.7.16, and all graph neural network components are built using DGL 2.0.0. Experiments are conducted on a single NVIDIA GeForce RTX 3090 GPU. For baseline comparisons, we use the reported results from Chen et al. (2024b) when available; otherwise, we reproduce the results using their official implementations and perform a grid search to optimize hyperparameters. All projectors and the classifier introduced in Section 4 are implemented as a two-layer MLP, both with 64-dimensional hidden representations. Model training is performed using the Adam optimizer (Kingma & Ba, 2014) with an initial learning rate of 0.001. We conduct a grid search for key hyperparameters: $\lambda_{cluster}$ and $\lambda_{pl}$ are selected from $\{5 \times 10^{-4}, 10^{-3}, 10^{-1}, 1\}$, and the batch size from $\{32, 64, 128\}$. The final hyperparameter settings for each dataset are detailed in Appendix A.

**Evaluation Protocol.** To evaluate our method under limited supervision, we follow the settings in Chen et al. (2024b) and use a low training ratio across all datasets. Specifically, for relatively larger datasets like T-Social, we set the training ratio to 0.01%. For other smaller datasets, we set the training ratio to 1%. The remaining data is divided into validation and testing sets in a 1:2 ratio. To comprehensively assess the model's performance, we employ metrics including AUROC, AUPRC, and F1-macro, which are particularly suitable for mitigating the effects of class imbalance. Higher values indicate better performance. Following the baseline approaches (Chen et al., 2024b), we select the best F1-macro on the validation set and use the corresponding threshold for evaluation on the test set. All reported results are averaged over five independent runs with different random seeds, and we report the mean and standard deviation.

## C ADDITIONAL EXPERIMENTS

### C.1 ANALYSIS OF HYPER-PARAMETERS

To analyse the effect of hyperparameters $\lambda_{pl}$ and $\lambda_{cluster}$, we conduct experiments on five datasets and take all three metrics to evaluate the performance. The results are summarized in Fig.4. The AUROC on Weibo is relatively stable across hyperparameter settings, while F1-Macro and AUPRC are more sensitive. Notably, $\lambda_{cluster}$ significantly affects performance when exceeding $10^{-3}$, suggesting that smaller values are preferable on Weibo. A similar trend is observed in Amazon, where a high $\lambda_{cluster}$ degrades performance, but a larger $\lambda_{pl}$ consistently improves all metrics. Notably, AUROC and F1-Macro/AUPRC require a trade-off with different emphases, highlighting the need for evaluation based on specific application demands. For T-Finance, better performance is achieved with a smaller $\lambda_{pl}$ and a larger $\lambda_{cluster}$, indicating that different datasets require different loss weightings. In these three datasets, performance generally improves as $\lambda_{pl}$ increases. Since the three datasets are relatively easier to train and yield higher-quality pseudo-labels, increasing the corresponding weight is reasonable.

### C.2 PERFORMANCES UNDER DIFFERENT LABEL RATIOS

To further evaluate the robustness of our framework under varying label ratios, we conduct a series of experiments on the Weibo and Amazon datasets. The results are presented in Fig.5 and Fig.6. Our method demonstrates strong performance across a wide range of label ratios on both datasets, particularly excelling in low-label scenarios. On the Weibo dataset, our method achieves consistent and significant improvements in AUROC and AUPRC across all label ratios, while also maintaining competitive performance in terms of F1-Macro. For the Amazon dataset, different methods excel in different metrics. For instance, BWGNN achieves the highest F1-Macro across most label ratios, yet its AUROC and AUPRC scores are noticeably lower than those of other methods. In contrast, our method delivers stable improvements over the baselines across all evaluation metrics. Overall, the results in Fig.5 and Fig.6 validate the scalability and effectiveness of our framework under varying

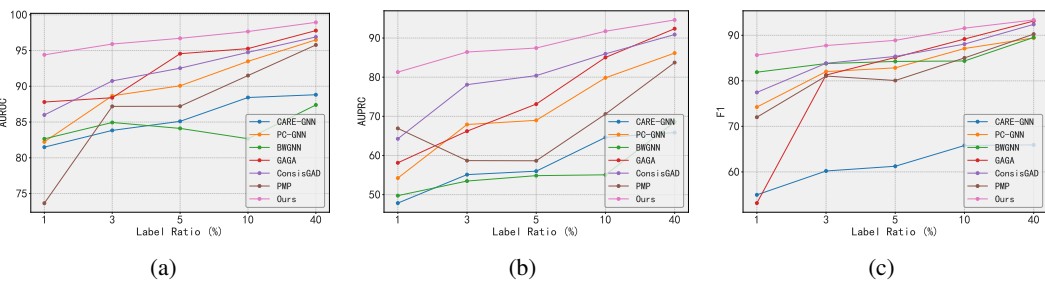

Figure 5: Performance comparison of different methods on the Weibo dataset across varying training ratios

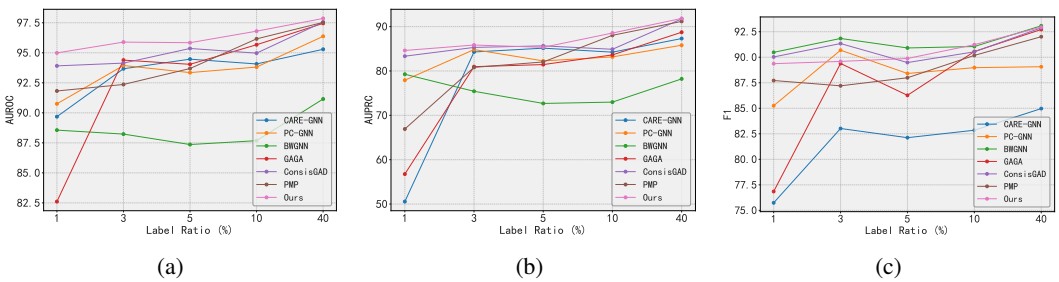

Figure 6: Performance comparison of different methods on the Amazon dataset across varying training ratios

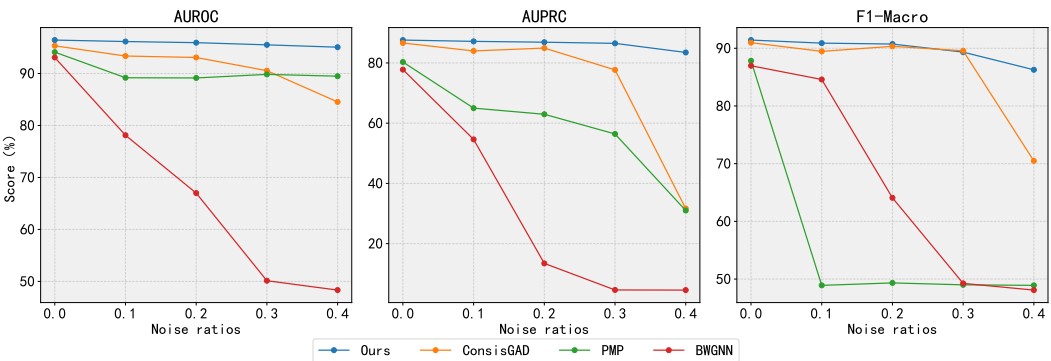

Figure 7: Performance on T-Finace dataset with different ratio of noisy labels.

proportions of labeled data. Notably, our method provides substantial gains in low-label settings and maintains competitive performance when more labeled data is available.

## C.3    ANALYSIS OF ROBUSTNESS

To evaluate the robustness of our proposed method, we simulate real-world scenarios where training data is subject to label noise attacks. Using the T-Finance dataset as an example, we start with only a small fraction of labeled data (1% of the dataset) and randomly flip a portion of the training labels to create noisy training sets. Keeping all other experimental settings unchanged, we progressively increase the label noise ratio to 0.1, 0.2, 0.3, and 0.4 to assess the performance of our method and various baseline models under different noise levels. The results are shown in Fig.7. As illustrated, our method consistently demonstrates superior robustness on the T-Finance dataset compared to the baselines. Although all models experience performance degradation as the noise ratio increases, the decline in our method is significantly more moderate. For example, in terms of AUROC, the performance drop remains below 2% even when 40% of the labels are corrupted. While the decline is relatively larger on the AUPRC metric, our method still outperforms all baselines

by a notable margin. This enhanced robustness is attributed to the collaborative multi-module architecture proposed in this work. In addition to learning a binary classifier from the limited labeled samples, our approach incorporates clustering and pseudo-labeling mechanisms, effectively exploiting the underlying structure in the large-scale unlabeled data. This enables the model to enhance its representation learning and noise tolerance capabilities.

