# OpenReview forum: "Coarse to Fine: Uncovering Node Diversity in Weakly Supervised Graph Anomaly Detection"
_ICLR.cc/2026/Conference — Submitted to ICLR 2026_

### Official Review · Reviewer_nzaS · 2025-10-26

**Soundness:** 2
**Presentation:** 3
**Contribution:** 2
**Rating:** 2
**Confidence:** 4

**Summary:**

This paper addresses the problem of weakly supervised graph anomaly detection, presenting a method that is clearly written and logically organized. The authors conduct several experiments to compare their approach with existing methods, demonstrating its effectiveness across different settings. While the paper introduces a unified gating classifier as a key technical component, the details and rationale behind this design are not fully explained, and the performance improvements over strong baselines are often marginal. Additionally, some aspects, such as the use of pseudo-labels and the impact of gating, are common or limited in their contribution.

**Strengths:**

1. This paper examines an interesting problem in weakly supervised graph anomaly detection, which has been extensively studied in the literature by numerous works.
2. The paper is well written and logically organized, making it easy to understand.
3. Several distinct experiments are conducted to compare the proposed method with existing approaches.

**Weaknesses:**

1.	The paper implies Figure 1b is problematic, but this issue is not explained. Does your method actually achieve a better decision boundary in this example?
2.	Figure 2 is not informative. It contains detailed icons of examples but does not clearly illustrate the technical designs, except for general labels such as clustering, loss symbols, and the overall training procedure.
3.	The technical design of the unified gating classifier is unclear in Section 4.1. The section does not explain what aspects are unified, what "gating" means, or its purpose. There are no technical details provided about this. Additionally, the rationale behind the gate vector design is unclear.
4.	The concept of pseudo-labeling mentioned at line 293 is common in the literature.
5.	The performance improvement on AUROC and Macro F1, especially as shown in Table 3, is often marginal compared to the best baseline.
6.	Gating itself does not significantly improve performance, as shown in Tables 4 and 5, yet it is presented as a major contribution of the work.

**Questions:**

Please see the six issues above.

---

### Official Review · Reviewer_iUpA · 2025-10-30

**Soundness:** 2
**Presentation:** 2
**Contribution:** 2
**Rating:** 4
**Confidence:** 3

**Summary:**

This paper addresses weakly supervised graph anomaly detection by arguing that binary classification overlooks the fine-grained diversity among normal and anomalous nodes. The authors propose a framework combining a frequency-aware gating module to adaptively leverage node-centric and neighborhood information, and a classifier-clustering synergy mechanism that iteratively maintains high-confidence sets to refine anomaly predictions. Experiments on seven public datasets show consistent improvements over existing methods.

**Strengths:**

1. Well-written and easy to follow
2. Strong experimental results demonstrates the effectiveness of the proposed method

**Weaknesses:**

1. Novelty is limited: This paper propsed a model consist of many different part and a complex optimize objective. Unfortunately, these efforts seems to be a combination of existing method.

2. Lack of qualitative validation: Although the paper claims to uncover “fine-grained anomaly subcategories,” there are no real semantic clusters or anomaly types verified in experiments.

3. Computational complexity & scalability not discussed sufficiently: The model applies clustering frequently and maintains dynamic sets, which could be costly for very large graphs.

**Questions:**

see weakness

---

### Official Review · Reviewer_TapA · 2025-11-02

**Soundness:** 2
**Presentation:** 2
**Contribution:** 2
**Rating:** 4
**Confidence:** 4

**Summary:**

The paper proposes a coarse-to-fine graph anomaly detection framework designed to capture node diversity under weak supervision. It introduces a frequency-aware gating module and a classifier–clustering synergy mechanism to balance contextual and structural information and discover fine-grained subcategories. Experimental results show consistent improvements across several benchmarks.

**Strengths:**

1. The paper is generally well-organised and easy to follow.
2. The paper show competitive performance on the selected datasets.

**Weaknesses:**

1. One main claim of the paper is that intra-anomaly separability relates to GAD performance. However, this is not discussed thoroughly in the paper. This is important for readers to understand the motivation of the paper. Also, structural and contextual anomalies seem to be important motivations of the design, but these are from synthetic dataset injections for creating datasets. They may not be sufficiently reflective of real-world abnormality.

2. Following the previous question, if the object remain binary anomaly scoring, as long as the overall GAD performance is acceptable, why would skipping subclass label prediction be a limitation? In my best understanding, no anomaly subclass label datasets are used for evaluation.

3. In the experiments, are there any evaluations on subclass performance? Also, what does unknown anomaly refer to? Does it refer to any unlabeled anomalies or those novel types of anomalies that have very different distributions from the training ones? f so, controlled experiments such as GDN-AugAN (Zhou et al., 2023) and NSReg (Wang et al., 2025) using datasets with ground-truth anomaly subclass labels should be conducted. If it’s closed-set GAD, then it would be good to let the authors know whether the improvement reflects better fitting or generalization.

4. The method improves from pseudo-label supervision that baselines do not use. Pseudo-labelling can generally improve the performance of any model, which makes it less clear to assess the dominant cause of the performance improvement.

5. The discussion on the limitations of supervised methods in the related work section is not clear. Related works on supervised methods (i.e., methods leveraging labels) are incomplete. To name a few: Graph-GDN (Ding et al., 2021), AMNet (Chai, 2022), ARC (Liu et al., 2024), and AnomalyGMF (Qiao et al., 2025).

6. For Figure 1(b), it would be better if illustrative plots are drawn from the actual latent space of some trained detector to better show the effect.

**Questions:**

Please refer to my weaknesses.

---

### Official Review · Reviewer_fV8X · 2025-11-03

**Soundness:** 2
**Presentation:** 3
**Contribution:** 2
**Rating:** 4
**Confidence:** 3

**Summary:**

This paper proposes a framework that uncovers anomalies in graphs using a coarse-to-fine strategy. The proposed method performs coarse anomaly localization and then refines detection using a fine-grained scoring mechanism. Experimental results show improved performance over baselines on public benchmark datasets.

**Strengths:**

S1. This paper tackles an important problem and offers a coarse-to-fine strategy for graph anomaly detection.

S2. It reports improved results for graph anomaly detection over several baselines across multiple datasets.

S3. The paper is generally well-structured and easy to follow.

**Weaknesses:**

W1. This paper offers limited novelty, as the coarse-to-fine idea and its components closely resemble existing multi-stage anomaly detection approaches.

W2. It lacks convincing evidence that the coarse-to-fine design is superior to strong single-stage baselines; ablation studies are somewhat insufficient.

W3. The evaluation depth is limited, with no analysis under varying anomaly ratios, different graph settings, or against more recent and competitive baselines.

**Questions:**

Q1. What concrete technical novelty does this method provide beyond prior multi-stage or hierarchical GAD approaches?

Q2. Can you provide ablations that disentangle the contribution of each stage to verify that the coarse-to-fine design, rather than model capacity or tuning, drives the improvements?

Q3. How does the method perform under more realistic and diverse settings (e.g., lower anomaly ratios, different graph types, and stronger recent baselines) to validate robustness and practical applicability?

---

### Meta-Review · Area_Chair_54zW · 2026-01-03

**Summary:**

This paper proposes a coarse-to-fine framework for weakly supervised graph anomaly detection, combining a frequency-aware gating module with a classifier-clustering synergy to model node diversity beyond binary anomaly labels.

While the problem setting is relevant and the paper is generally well written, the overall contribution is somewhat limited. Across reviews, the method is widely viewed as an aggregation of existing ideas (e.g., gating, pseudo-labeling, and clustering) without delivering sufficiently clear or novel technical insights. Although empirical results show modest improvements over selected baselines, these gains are often marginal and not convincingly attributed to the proposed coarse-to-fine design itself. In the absence of an author rebuttal, the reviewers’ concerns regarding novelty, motivation, and evaluation remain unresolved.

**Reviewer Concerns:**

A consistent concern across reviewers is limited novelty. Reviewers TapA, iUpA, and fV8X all note that the core components closely resemble prior multi-stage or self-training approaches in graph anomaly detection, and that the paper does not clearly articulate what fundamentally new insight is introduced beyond combining known techniques. In particular, the motivation that fine-grained intra-anomaly diversity is critical for improving GAD performance is not theoretically or empirically substantiated, as pointed by TapA.

The conceptual justification and evaluation of node subcategories are also weak. While the paper claims to uncover fine-grained anomaly subtypes, no datasets with ground-truth anomaly subclasses are used, and no subclass-level or qualitative validation is provided. As pointed out by TapA, it remains unclear whether the task is open-set or closed-set GAD, and whether the reported improvements reflect better generalization or simply stronger fitting under pseudo-label supervision.

The evaluation protocol is insufficiently convincing. Multiple reviewers highlight the lack of robustness analysis under more realistic conditions, such as varying anomaly ratios, different graph types, or stronger and more recent baselines.

Finally, method clarity and completeness are concerns. Reviewers question the clarity of the unified gating design, the marginal contribution of gating based on ablations, and the absence of discussion on computational complexity and scalability, particularly given the repeated clustering and dynamic set maintenance.

The above concerns are outstanding due to the absence of an author’s rebuttal.

**Reviewer Scores:**

Three reviewers (fV8X, TapA, iUpA) rated the paper 4, explicitly mentioning limited contribution and insufficient evidence for the claimed advantages. One reviewer (nzaS) gave a 2, emphasizing unclear technical design, marginal gains over strong baselines, and overstated contributions. Given that no rebuttal was provided to address these shared concerns, it is unlikely that reviewers would have increased their scores had further discussion been possible; if anything, the concerns raised by TapA and nzaS present a downward pressure on the overall assessment.

---

### Decision · Program_Chairs · 2026-01-26

Reject